# Optimizing NiFe-Modified Graphite for Enhanced Catalytic Performance in Alkaline Water Electrolysis: Influence of Substrate Geometry and Catalyst Loading

**DOI:** 10.3390/molecules29194755

**Published:** 2024-10-08

**Authors:** Mateusz Kuczyński, Tomasz Mikołajczyk, Bogusław Pierożyński, Jakub Karczewski

**Affiliations:** 1Department of Chemistry, Faculty of Agriculture and Forestry, University of Warmia and Mazury in Olsztyn, Łódzki Square 4, 10-727 Olsztyn, Poland; mateusz.kuczynski@uwm.edu.pl (M.K.); tomasz.mikolajczyk@uwm.edu.pl (T.M.); 2Institute of Nanotechnology and Materials Engineering, Faculty of Applied Physics and Mathematics, Gdansk University of Technology, ul. G. Narutowicza 11/12, 80-233 Gdańsk, Poland; jakub.karczewski@pg.edu.pl

**Keywords:** HER, OER, NiFe, graphite, alkaline water electrolysis

## Abstract

The oxygen evolution reaction (OER) and the hydrogen evolution reaction (HER) are critical processes in water splitting, yet achieving efficient performance with minimal overpotential remains a significant challenge. Although NiFe-based catalysts are widely studied, their performance can be further enhanced by optimizing the interaction between the catalyst and the substrate. Here, we present a detailed investigation of NiFe-modified graphite electrodes, comparing the effects of compressed and expanded graphite substrates on catalytic performance. Our study reveals that substrate geometry plays a pivotal role in catalyst distribution and activity, with expanded graphite facilitating more effective electron transfer and active site utilization. Additionally, we observe that increasing the NiFe loading leads to only modest gains in performance, due to catalyst agglomeration at higher loadings. The optimized NiFe–graphite composites exhibit superior stability and catalytic activity, achieving lower overpotentials and higher current densities, making them promising candidates for sustainable hydrogen production via alkaline electrolysis.

## 1. Introduction

The quest for sustainable and efficient energy sources has led to a significant focus on alkaline water splitting, a promising method for clean and renewable hydrogen production. Central to this process is the role of electrocatalysts, among which NiFe-based catalysts have emerged as a frontrunner. Their deposition on conductive substrates, such as expanded graphite and compressed graphite, enhances the performance of graphite for both hydrogen and oxygen evolution reactions (HER and OER, respectively), which are integral to the phenomenon of water electrolysis [1,2,3,4,5,6,7,8].

NiFe catalysts are distinguished by their unique structural and electronic properties. Combined investigations by means of X-ray absorption near-edge structure (XANES) and extended X-ray absorption fine structure (EXAFS) techniques have revealed square planar molecular structures in these catalysts, this being especially advantageous for their catalytic activity [9,10].

The electrochemical oxygen evolution and surface redox chemistry of NiFe layered double hydroxides (LDHs) have also been studied. The results produced by cyclic and linear sweep voltammetry showed that NiFe LDHs exhibited excellent catalytic activity under alkaline conditions, significantly outperforming Fe-free hydroxide analogs. Their low overpotentials for the OER indicate superior catalytic efficiency (with a current density of 10 mA cm^−2^ for an overpotential of 348 mV) [11].

The X-ray diffraction (XRD) analysis of NiFe nanoparticles (NPs) further elucidates their structural features, which are essential for understanding their catalytic behavior. The unique composition and formation of these NPs, as indicated by the presence of γ-Fe_2_O_3_ and the absence of nickel oxide diffraction peaks, suggest a specific structural configuration, leading to enhanced OER catalytic activity [12].

A key challenge in improving NiFe catalysts is the optimization of their interaction with the supporting substrate, which plays a crucial role in determining the overall catalytic performance. Substrate geometry and surface characteristics significantly influence the distribution of the catalyst, the accessibility of active sites, and the efficiency of electron transfer. While extensive research has focused on optimizing the composition and synthesis of NiFe catalysts, less attention has been paid to the physical properties of the substrates used and how these influence catalytic performance. Studies have shown that factors such as substrate morphology can dramatically affect catalytic activity, yet a systematic investigation into these effects, particularly in the context of NiFe-modified graphite, is missing.

This study focuses on the exploration of NiFe catalysts, specifically their deposition and behavior toward HER and OER processes of the expanded graphite/NiFe and compressed graphite/NiFe catalyst composites produced in this way. The choice of graphite as a substrate stems from its excellent conductivity, mechanical stability, and cost-effectiveness, making it an ideal candidate for large-scale applications [13]. Our investigation delves into the synthesis and characterization of NiFe catalysts on these substrates, aiming to understand the interaction between the catalyst and the substrate and how this impacts the overall efficiency of electrochemical water-splitting.

The novelty of this work lies in its dual focus on substrate geometry and catalyst loading, which together offer a more comprehensive understanding of how to optimize the content of NiFe-based catalysts for practical applications in hydrogen production. By addressing these critical factors, our study contributes to broader efforts to develop more efficient and scalable water electrolysis systems, aligning with the ongoing research aimed at overcoming the bottlenecks associated with current NiFe oxyhydroxide catalysts.

## 2. Materials and Methods

### 2.1. Solutions and Chemical Reagents

All solutions were prepared using ultra-pure water with a resistivity of 18.2 MΩ cm, obtained from a Spring 30 s water purification system (Hydrolab, Gdańsk, Poland). First, 0.1 and 1.0 M NaOH supporting solutions were prepared using sodium hydroxide pellets (99.9%, POCH, Gliwice, Poland). Then, a palladium reversible hydrogen electrode (RHE) was charged in a 0.5 M H_2_SO_4_ solution prepared from concentrated sulfuric acid (98%, Merck, Darmstadt, Germany).

### 2.2. Electrodes and Electrochemical Cell

Electrochemical experiments were conducted in a standard electrochemical cell made of Pyrex glass. The cell comprised three electrodes: a graphite-based working electrode (WE), a Pd RHE (Pd wire, 99.99% purity, 1.0 mm diameter, Sigma-Aldrich, Poznań, Poland) acting as a reference electrode (RE), and a Pt counter electrode (CE; Pt wire, 99.99% purity, 1.0 mm diameter, Sigma-Aldrich). Each electrode was placed in a separate compartment. Before carrying out any experiments, atmospheric air was removed from the cell by bubbling with argon (Ar 5.0 grade, Linde Gas Poland, Olsztyn, Poland), and an argon gas flow was continuously maintained above the solutions throughout the experiments.

### 2.3. Expanded Graphite and Compressed Graphite Electrodes

The expanded graphite and compressed graphite (see details in Appendix A) plate electrodes, used as the working electrodes, had a size of 2 × 2 cm and corresponding thicknesses of 1.0 and 3.0 mm. Initially, they were ultrasonicated in acetone in order to remove any surface impurities that could otherwise be present.

The electrodes’ geometrical surface areas (GSA) for expanded graphite and compressed graphite were 8.6 and 9.8 cm^2^, respectively. All electrochemical parameters are reported based on these GSA estimations. However, results based on the electrochemically active surface area (ECSA) are also included to provide a more detailed understanding of their electrochemical behavior. The ECSA, estimated from a double-layer capacitance (*C*_dl_), offers a more accurate measure of the active sites available for electrochemical reactions. The values of ECSA for the graphite electrodes are provided in the Appendix A.

Electrodeposition of the NiFe catalyst on the graphite electrodes was carried out with four different loadings: 0.5 mg, 2.5 mg, 5.0 mg, and 7.5 mg of NiFe per 1 cm^2^. The conditions for the NiFe electrodeposition and composition of produced deposits are reported in Appendix A.

Procedures for setting up the laboratory equipment and materials were consistent with those previously described in other works from this laboratory [14,15,16].

### 2.4. Experimental Methodology

Electrochemical measurements:

Electrochemical measurements were conducted, utilizing an AUTM 204 + FRA 32M Multi-Autolab potentiostat/galvanostat system in a 0.1 M NaOH solution as the supporting electrolyte. The choice of working solution enabled the best comparisons of the results with the existing literature.

Quasi-steady-state Tafel polarization: This technique was used to study the kinetics of the hydrogen evolution reaction (HER) and oxygen evolution reaction (OER) on the NiFe-modified expanded graphite and compressed graphite electrodes. Tafel polarization experiments were conducted at a scan rate of 0.5 mV s^−1^.

Electrochemical impedance spectroscopy (EIS): EIS measurements were conducted to investigate the electrode kinetics and charge transfer processes. The frequency range for EIS was typically between 1.0 × 10^5^ and 0.5 × 10^−1^ Hz, with an output signal amplitude of 5 mV.

Cyclic voltammetry (CV): CV measurements were performed to assess the electrochemical behavior and stability of the electrodes. Three sweeps were always carried out over the potential span of −1.0 to 1.8 V vs. RHE at a scan rate of 5 and 50 mV s^−1^.

Surface characterization:

A combination of advanced spectroscopy techniques was employed to comprehensively characterize the surface properties of the produced electrodes.

Scanning electron microscopy/Energy-dispersive X-ray (SEM/EDX) surface spectroscopy characterization was performed for all examined samples, including the NiFe-modified expanded graphite and compressed graphite. These analyses were carried out using a Quanta FEG 250 (FEI Company, Brno, Czech Republic) microscope equipped with Bruker XFlash 6010 EDX (Berlin, Germany) instrumentation (with a resolution of 125 eV).

X-ray diffraction (XRD) analysis: The phase composition of the bulk samples was analyzed using an X-ray diffraction method by means of the X’Pert Pro MPD Philips (Panalytical, Almelo, Netherlands) diffractometer, using Cu Kα (1.542 Å) radiation at room temperature.

Atomic force microscopy (AFM) measurements were conducted using a Nanosurf CoreAFM (Liestal, Switzerland) system to characterize the surface topography and roughness of both unmodified and NiFe-modified graphite electrodes. For each sample, multiple areas were scanned to ensure representative results. The scan sizes were set to 1 × 1, 5 × 5, and 10 × 10 μm. The images were processed using the Nanosurf software v. 3.8 to analyze the surface roughness parameters.

Software and data analysis:

The electrochemical instruments were controlled by means of Nova 2.1 software (Metrohm Autolab B.V., Opacz-Kolonia, Poland). All impedance data were analyzed using the ZView 2.9 software package (Scribner Associates, Inc., Berwyn, PA, USA). The impedance spectra were fitted by the LEVM 6 program, a complex, non-linear, least-squares immittance fitting tool [17].

XT microscope control software (v. 2.3) ran the SEM microscope for image capture and analysis. Meanwhile, Bruker’s QUANTAX EDS has been integrated into the ESPRIT software (v. 2.1) suite for qualitative and quantitative elemental analyses.

## 3. Results and Discussion

### 3.1. Surface Characterization

#### 3.1.1. SEM/EDS Analysis

Figure 1a–e and Figure 2a–e of this study reveal the structural characteristics of the NiFe alloy deposited on the expanded graphite and compressed graphite. The EDS images show that the NiFe structures on these substrates exhibit a regular and homogenous morphology for all NiFe loadings (see Figure 1b and Figure 2b). However, despite the overall uniformity, there were instances where the carbon element was visibly discernible within the structure, indicating areas of lesser NiFe coverage or integration.

The analysis determined that the grain size of the deposited NiFe alloy varied between 40 and 60 nm. Additionally, there were noticeable agglomerates of NiFe particles, with some reaching sizes of up to 150 nm. The weight of NiFe alloy for each loading amount (0.5, 2.5, 5.0, and 7.5 mg of NiFe per cm^2^) was confirmed by the weighting method, with an error of about 4.5%. Furthermore, some variations were observed in the elemental composition percentages in the SEM/EDX analysis of the NiFe alloy content, depending on the deposition time and the graphite substrate (for more details, see Appendix A).

#### 3.1.2. AFM Analysis

Atomic force microscopy (AFM) was employed to examine the topographical characteristics of NiFe-modified graphite electrodes. The analysis focused on an assessment of the surface roughness and distribution of the NiFe catalyst on the substrates. The AFM images (Figure 3a,b) revealed a uniform deposit of NiFe particles with visible agglomeration and higher catalyst loadings contributing to an increased active surface area, this being essential for electrocatalytic processes. Furthermore, quantitative data on surface roughness parameters were obtained (Appendix A) linking the microstructural features to the electrochemical performance, as observed by cyclic voltammetry and impedance spectroscopy studies. These findings highlight the correlation between the nanoscale surface attributes and the overall efficiency of the electrodes in electrolysis applications.

#### 3.1.3. XRD Analysis

Figure 4 presents the X-ray diffractograms for samples deposited on respectively expanded graphite (Figure 4a) and compressed graphite (Figure 4b). In both cases, a graphite substrate with a hexagonal structure is clearly visible. In the case of compressed graphite, the structure is more oriented and only the planes of the 002 family are visible. The presence of the NiFe phase can be observed above surface concentrations of 2.5 mg cm^−2^, and the amount of this phase increases with rising NiFe concentration. The NiFe phase crystallizes in a regular Pm-3m structure, and the significant broadening of the observed peaks indicates its nanostructural character.

### 3.2. Electrochemical Characterizations

#### 3.2.1. Cyclic Voltammetry

The cyclic voltammetry (CV) analysis detailed in Figure 5a,b of this study provides a comparative insight into the electrochemical behavior of all the examined electrodes (expanded graphite, compressed graphite, and all their NiFe modifications) in contact with a 0.1 M NaOH solution. The latter was achieved through CV sweeps across the potential range of −1.0 to 1.8 V vs. RHE, with a scan rate of 50 mV s^−1^, focusing on the last cycles to assess the voltammogram’s stability and reproducibility. The deposition of the NiFe alloy onto the graphite substrates led to a marked improvement in the catalysis of both the HER and OER processes. However, an increase in the NiFe loading did not proportionately enhance the catalytic properties. For the expanded graphite, the shape of the CV profiles suggested that while there is an initial improvement in catalytic behavior with the introduction of the NiFe alloy, further raising the NiFe loading value does not lead to a reasonable increase in the catalytic performance. This near-plateau in observed current density (Figure 5a) could most likely be attributed to the process of agglomeration of the catalyst particles at higher loadings, which may then inhibit the electron transfer process and active site utilization [18,19].

In contrast to the expanded graphite, the CV profiles for the compressed graphite electrodes displayed an inverse relationship between the NiFe loading and the catalytic performance. The obtained data suggest that the lowest NiFe loading yields the best catalytic behavior for the compressed graphite, outperforming those of composite electrodes with higher NiFe loadings. This unexpected trend could imply a more homogeneous distribution and utilization of active sites at lower catalyst loadings on this material, or else it might reflect differences in the interaction between the NiFe particles and the compressed graphite surface. Such a trend underscores the complex nature of catalyst–substrate interactions and the importance of optimizing loading levels for different substrate geometries to achieve the best possible electrochemical performance. Moreover, the characteristics of the NiFe oxidation/reduction peaks in the CV profiles were in line with the phenomena regarding the formation/transition of hydroxide and oxyhydroxide phases described in previous research from this laboratory [13].

Additionally, cyclic voltammetry measurements were used to determine the working electrodes’ double-layer capacitance, which is strongly correlated to the electrochemically active surface area (ECSA). In order to accurately assess the double-layer capacitance (*C*_dl_) of electrodes, CV measurements were conducted around the potential of 0.7 V vs. RHE, within a potential sweep range of ±0.10 V. This specific potential range was chosen to ensure operations within the non-Faradaic CV region, focusing on capturing the capacitive behavior of the electrode–electrolyte interface without interference from redox processes. The sweep rates for these CV measurements varied between 5 and 100 mV s^−1^, allowing for an examination of the current response across a broad range of kinetic conditions. The *C*_dl_ was determined by analyzing the relationship between the capacitive current and the sweep rate. In the absence of Faradaic processes, the capacitive current at a given potential is expected to be directly proportional to the sweep rate, following the equation I = v*C*, where I is the current, v is the sweep rate, and *C* is the capacitance. The slope obtained from plotting the capacitive current against the sweep rate, as illustrated in Appendix A, directly quantifies the *C*_dl_ parameter [20,21]. The values derived from this analysis are presented in Table 1.

The change in the *C*_dl_ in terms of the function of the NiFe loading on expanded graphite and compressed graphite shows a clear trend. For the expanded graphite, as the loading of NiFe increases from 0 to 7.5 mg cm^−2^, the *C*_dl_ significantly decreases from 9000 µF to 2500 µF. This phenomenon is likely attributable to the NiFe catalyst layer’s tendency to cover the intrinsic microstructure of the expanded graphite, potentially obscuring some of its active sites.

On the compressed graphite, however, the trend is not as straightforward. Starting with an unmodified electrode with a *C*_dl_ of 2900 µF, there is an initial capacitance increase to 22,400 µF at a NiFe loading of 0.5 mg cm^−2^, which might indicate that small amounts of NiFe catalyst enhance the electroactive surface area or affect the surface roughness in a manner that considerably increases the capacitance. Beyond an initial capacitance increase at a low NiFe loading, the *C*_dl_ value demonstrates a decreasing trend with further increments of the NiFe catalyst on the compressed graphite. This reduction in the capacitance suggests that beyond a certain level, additional NiFe deposits may lead to a reduction in the electrode’s available surface area for charge accumulation, potentially due to particle agglomeration or changes in the surface microstructure, as evidenced by the SEM analysis (see Figure 1 and Figure 2) [22,23].

As presented in Appendix A, the AFM analysis revealed that the compressed graphite had a more developed surface area than that of the expanded graphite, thus implying its higher potential for electroactivity. However, the electrochemical tests, such as cyclic voltammetry or AC impedance, indicated that the expanded graphite actually exhibited higher activity. This apparent contradiction could be explained by considering the active sites’ nature and activity. The AFM results imply that while the compressed graphite has a larger surface area (Appendix A), not all of its area may consist of electrochemically active sites. Despite having a less developed surface, the expanded graphite presents a higher proportion of accessible active sites. However, when NiFe is deposited at a low loading level on the graphite substrates, the *C*_dl_ parameter for the compressed graphite shows a significant increase, opposite to the expanded graphite, based on the CV measurements. This suggests that the NiFe catalyst effectively utilizes the larger surface area of the compressed graphite at its low loadings, enhancing the electroactivity by interacting with the existing active sites and potentially creating new ones by the deposition of NiFe. However, as the NiFe loading increases, the subsequent decrease in the *C*_dl_ might indicate that the NiFe catalyst starts agglomerating (see Figure 1 and Figure 2) or commences blocking the substrate’s pores, thereby reducing the effective substrate’s surface area. This explanation aligns with the observed electrochemical behavior, whereby an initial deposition of NiFe improves the catalyst activity, as indicated by the increased *C*_dl_ values; further NiFe deposition leads to deterioration of the catalytic performance, which is likely due to overcrowding of the catalyst and a significant reduction in the number of accessible active sites. This subtly led to the use of the dimensional geometrical area as a means to better characterize the materials, thus enabling their more direct comparison. This approach takes into consideration the practical application of these materials in real-scale alkaline water electrolyzers.

#### 3.2.2. AC Impedance—HER

Figure 6a and Table 2 of this article illustrate the selected impedance spectroscopy results (the full potential range is presented in Appendix A) for electrodes modified with NiFe on the expanded graphite and the compressed graphite base material electrodes, all tested in 0.1 M of NaOH solution. The electrochemical parameters, such as charge transfer resistance (*R*_ct_), porosity resistance for reaction intermediates (*R*_p_), double-layer capacitance (*C*_dl_), and pseudo-capacitance (*C*_p_), were derived from the data using equivalent circuit models, as depicted in Figure 6b–d.

For unmodified expanded graphite electrodes, the impedance measurements at low overpotentials (from −50 to −400 mV vs. RHE) demonstrated a single depressed semicircle at high frequencies, indicative of a porosity response, and a linear region at medium to low frequencies that corresponded to capacitive behavior. As the probing potential was displaced to more negative values, specifically, to the range between −450 and −700 mV vs. RHE, an additional semicircle appeared in the impedance spectrum, signifying the onset of the HER process at medium and low frequencies. For the expanded graphite electrodes, the *R*_p_ and *C*_p_ parameters were largely found to be potential-independent in reference to the intrinsic characteristics of the electrode’s porous entity. However, a significant increase in the *C*_dl_ parameter was observed from 7999 to 12,815 µF cm^−2,^ as the cathodic overpotential increased from 50 to 700 mV. The latter may indicate that as the overpotential rises, more of the expanded graphite’s surface area becomes electrochemically active, which could be due to the electrode’s relatively poor catalytic properties and highly electrochemically non-uniform surface. Additionally, the *R*_ct_ parameter associated with HER reactivity, which was monitored in the potential range of −450 to −700 mV, showed a notable decrease from 396.2 to 24.5 Ω cm^2^. This substantial reduction in the charge transfer resistance is characteristic of the kinetically controlled process.

Then, upon NiFe alloy deposition, the expanded graphite’s “porosity” response, indicated by the semicircle corresponding to the *R*_p_ and *C*_p_ parameters, was no longer discernible at more negative potentials than −150 mV. This change is attributed to the extended formation and accumulation of hydrogen (H_2_) bubbles during the HER process.

Additionally, the introduction of the NiFe catalyst onto the expanded graphite significantly enhanced the HER kinetics, as evidenced by the onset of the HER process at lower overpotentials (Figure 5a). Also, catalytic graphite modification caused a radical reduction in the *R*_ct_ value by approximately 57 times for a NiFe loading of 0.5 mg cm^−2^, as compared to the *R*_ct_ recorded for the unmodified expanded graphite at a potential of −450 mV. This reduction in the *R*_ct_ indicates a significantly improved rate of electron transfer during the HER, highlighting the effectiveness of the NiFe catalyst in promoting the hydrogen evolution process, which is also correlated with an increase in the current density observed in the HER region for the CV measurements. Also, for the NiFe-modified electrodes under increasing cathodic overpotentials (from −50 to −450 mV), a substantial reduction in the *R*_ct_ and *C*_dl_ parameter values was observed (see Table 2). Hence, the *R*_ct_ parameter decreased from 123.1 Ω cm^2^ to 6.1 Ω cm^2^ within the tested potential range, indicating a significant enhancement of the HER kinetics at the electrode surface, while a decrease in the *C*_dl_ parameter from 13,277 to 3474 µF cm^−2^ was most likely caused by the evolution of hydrogen gas at the electrode surface, which could considerably obstruct its electroactive areas, thus leading to a reduced effective surface area.

In our exploration of the NiFe catalyst loadings on expanded graphite electrodes for enhanced electrocatalytic applications, our investigation revealed a subtle relationship between the amount of catalyst and the observed catalytic performance. Initial loading of the NiFe (0.5 mg cm^−2^) on the expanded graphite electrodes resulted in a marked improvement in the electrochemical behavior, as evidenced by the reduction in the charge transfer resistance and an increase in the double-layer capacitance (see Figure 6a and Table 2). However, as the catalyst loading was increased to 2.5, 5.0, and 7.5 mg cm^−2^, the anticipated continuation of the performance enhancement did not materialize. Instead, a decline in the catalytic efficiency of the composites was observed; namely, the *R*_ct_ parameter increased by 1.01, 1.45, and 2.24 times for 2.5, 5.0, and 7.5 mg cm^−2^ of the NiFe loading, respectively. The above finding is in contrast to expectations based on the composites’ increase in the real surface area, as derived based on the AFM measurements. This actual surface area increase suggests greater textural complexity, due to the catalyst’s deposition, yet the electrochemically active surface area obtained from the *C*_dl_ measurements was considerably reduced.

Higher catalyst loadings likely lead to NiFe particle agglomeration. Although this increases the electrode’s textural complexity and the real surface area, it may reduce access to active sites, thus diminishing its effective electrochemical surface area. With increased NiFe loading, the surface chemistry modifications could significantly influence the adsorption-desorption kinetics of hydrogen. The interaction between pre-existing active sites on the graphite substrate and those introduced by the lowest NiFe catalyst loadings could eventually lead to an enhancement in the composite’s electrocatalytic activity.

In contrast to the behavior of the expanded graphite, the EIS analysis of the unmodified compressed graphite electrodes (see Figure 7 and Table 2 and Appendix A) exhibited two distinct semicircles across the potential range of −50 to −700 mV vs. RHE. The semicircle observed at high frequencies corresponded to the porosity of the electrode, reflecting the resistance and capacitance associated with the electrode’s porous structure. The values of the *R*_p_ and *C*_p_ parameters were relatively stable and were not influenced by the applied potential, with *R*_p_ showing fluctuations between 2.9 and 7.6 Ω cm^2^, and *C*_p_ values varying between 4066 and 9783 µF cm^−2^. The semicircle visible in the medium to low-frequency range was associated with HER reactivity. The *R*_ct_ parameter decreased (from 2162.0 to 140.0 Ω cm^2^) with increasing overpotential, suggesting an enhancement in the kinetics of electrochemical reactions at higher overpotentials. Meanwhile, the *C*_dl_ parameter exhibited a significant increase (2766 and 40,964 µF cm^−2^), indicating activation of the electrochemical surface area related to the rising overpotential.

It is important to highlight that while the expanded graphite did not show significant reactivity toward the HER until reaching the potential of −450 mV, the *R*_ct_ at this potential point was slightly lower than that observed for the compressed graphite, showing 1.25× enhanced catalytic activity. This disparity in the *R*_ct_ values further widened with the rising overpotential and eventually reached an approximately 2.11× reduction in favor of the expanded graphite at a potential of −700 mV. Despite its delayed reactivity, this lower *R*_ct_ value recorded on the expanded graphite could be due to its intrinsic structural and electrochemical properties, such as the efficient arrangement or exposure of catalytically active sites to the working solution. This nuanced behavior highlights the complex interplay between surface area expansion, active site distribution, and electrochemical activity, underscoring the finding that a larger surface area of compressed graphite does not directly translate into enhanced catalytic efficiency.

Then, upon modification with the NiFe alloy (0.5 mg cm^−2^), the compressed graphite electrodes demonstrated a significant improvement in their electrochemical properties. Notably, the *R*_ct_ values were approximately 86.67× lower at the potential of −450 mV for the NiFe-modified compressed graphite than for its unmodified counterpart. This substantial decrease in the *R*_ct_ underscores the effectiveness of the NiFe catalyst in facilitating the electron transfer process, thereby enhancing the overall electrocatalytic activity of the electrodes. Interestingly, after NiFe modification, no porosity response was detected by the AC impedance tool.

Similarly to the behavior observed in the unmodified compressed graphite, the NiFe-activated electrodes exhibited analogous trends in the *R*_ct_ and *C*_dl_ parameters upon rising overpotential. Moreover, analogously to the observations of the expanded graphite-based electrodes, the compressed graphite-based specimens also exhibited a decline in their catalytic efficiency with increasing NiFe loading levels. Specifically, a reduction in their efficiency by factors of 2.06, 6.70, and 8.98 for NiFe loadings of 2.5, 5.0, and 7.5 mg cm^−2^ was observed, respectively, at a potential of −50 mV, corresponding to 0.5 mg cm^−2^. This trend underscores the complex interplay between catalyst loading and electrode performance, where beyond a certain threshold, any additional catalyst does not translate to improved efficiency and may, in fact, hinder the catalytic process, due to such important factors as catalyst agglomeration, the blocking of active sites, or the detrimental effects of increased hydrogen bubble accumulation.

The relationship between the logarithmic inverse of the charge transfer resistance (−log *R*_ct_) and overpotential (*η*) was analyzed for kinetically controlled reactions across a potential range from −50 to −700 mV vs. RHE, with intervals of 50 mV. This analysis enabled the calculation of exchange current densities (*j*_0_) for the HER, leveraging the Butler–Volmer equation to correlate *j*_0_ with the *R*_ct_ parameter as the overpotential approached zero [14].

For expanded graphite and compressed graphite electrodes, including those modified with varying loadings of NiFe alloy (such as 0.5, 2.5, 5.0, and 7.5 mg cm^−2^), the calculated *j*_0_ values are presented in Table 3.

For an unmodified expanded graphite electrode, a baseline *j*_0_ value of 2.78 × 10^−7^ A cm^−2^ at low overpotential ranges signifies a small amount of inherent activity for the HER. The introduction of the NiFe catalyst markedly amplifies this activity, exhibiting the effectiveness of NiFe in promoting the hydrogen evolution process. However, an intriguing pattern emerges with increasing NiFe loadings; the *j*_0_ values at a low overpotential range (50 to 200 mV) decrease from 5.7 × 10^−5^ to 1.5 × 10^−5^ A cm^−2^ as the loading progresses from the lowest to the highest level. This trend suggests that additional NiFe does not translate to improved catalytic efficiency beyond a certain threshold, possibly due to factors such as catalyst agglomeration or the surface saturation of active sites.

The unmodified compressed graphite electrodes start with a higher baseline *j*_0_ of 3.12 × 10^−6^ A cm^−2^ at low overpotential ranges, compared to the expanded graphite, highlighting the differences in material properties and their effect on HER activity. Then, modification with NiFe (0.5 mg cm^−2^) particles significantly increases the *j*_0_ values to 1.7 × 10^−4^ and 4.3 × 10^−4^ A cm^−2^ for low and high overpotential ranges (50–200 and 200–500 mV), respectively. This enhancement underscores the potential of NiFe as an effective catalyst across different substrate materials. Similarly to the expanded graphite electrodes, the compressed graphite materials also exhibit a descending trend in the *j*_0_ values with increased loadings of NiFe, culminating in a decrease to about 1.2 × 10^−5^ and 2.6 × 10^−4^ A cm^−2^ for low and high overpotential ranges at the highest NiFe loading. This decline across both low and high overpotential ranges for compressed graphite electrodes further emphasizes the subtle balance between optimizing catalyst loading and achieving maximum catalytic efficiency.

#### 3.2.3. Polarization Technique—HER

The Tafel polarization plots for expanded graphite and compressed graphite electrodes, both unmodified and activated with various NiFe loadings (0.5, 2.5, 5.0, and 7.5 mg cm^−2^), are illustrated in Figure 8. The cathodic Tafel slopes (*b*_c_) and the exchange current densities (*j*_0_) for the HER are detailed in Table 3. For the NiFe-modified expanded graphite and compressed graphite samples, measurements were taken within the potential range of −50 to −200 mV vs. RHE. However, due to the delayed onset of hydrogen evolution on the unmodified graphite samples, a more negative potential range of −700 to −900 mV vs. RHE was selected for these electrodes. This shift in potential range for the onset of hydrogen evolution on unmodified graphite electrodes aligns with previous observations made with the EIS results.

The *j*_0_ values derived from the Tafel plots are in line with those calculated by means of the Butler–Volmer equation, indicating a significant enhancement in catalytic properties following NiFe modification. Additionally, these electrodes demonstrated a more positive onset potential for H_2_ evolution, compared to their unmodified counterparts (see Figure 9). The exchange current density values derived from this analysis are in good agreement with the literature values, reinforcing the validity of the observed trends [24,25].

The exchange current density (*j*_0_) is a fundamental parameter that provides deep insights into the intrinsic electrochemical activity of an electrode material, highlighting how readily the HER can be initiated without significant energy barriers. This parameter essentially quantifies the electrode’s efficiency in catalyzing the HER at a point where the overpotential is zero, making it a valuable indicator of the material’s catalytic capability under equilibrium conditions.

Despite the significant role of *j*_0_ in providing insights into the intrinsic electrochemical activity of electrode materials for the HER, it is frequently observed that many studies in the field prioritize parameters, such as the Tafel slope (*b*_c_), and operational metrics, such as the overpotential needed to reach a current density of 10 mA cm^−2^. This focus stems from the relative ease of measuring these operational standards, compared to calculating the *j*_0_ parameter, which demands in-depth electrochemical analysis incorporating the Butler–Volmer equation. By providing direct, application-relevant insights into electrode performance, operational metrics become invaluable for comparative and benchmarking purposes.

In order to align our study with common literature practices, especially regarding research emphasizing operational benchmarks, we also conducted these specific measurements in 1 M of NaOH solution (see Appendix A). The concentrations used in this work are widely recognized for their ability to offer a consistent and relevant benchmark for comparison, thus allowing for the direct analogy of the operational performance of our electrode materials against a wide array of existing research findings. By selecting both concentrations for these measurements, our study not only adheres to established experimental standards but also ensures a clearer and more impactful comparison of our results with those in a broader field. These operational metrics are presented in Table 3, thus ensuring a comprehensive overview of our electrodes’ performance in a context that maximizes both scientific rigor and relevance to ongoing research efforts.
molecules-29-04755-t003_Table 3Table 3HER kinetic parameters for the selected catalytic materials.Base MaterialCatalystLoading [mg cm^−2^]ElectrolyteEISTafelRefs.*j*_0_ [A cm^−2^]*j*_0_ [A cm^−2^]*b*_c_ [mV dec^−1^]*η*_(*j*=10 mA cm^−2^)_ [mV]Low *η*High *η*Unmodified expanded graphite None00.1 M NaOH2.78 × 10^−7^2.78 × 10^−7^1.97 × 10^−6^203-This article1.0 M NaOH--2.18 × 10^−6^163596Expanded graphiteNiFe0.50.1 M NaOH5.67 × 10^−5^2.37 × 10^−4^2.00 × 10^−4^137290This article1.0 M NaOH--1.40 × 10^−4^112200Expanded graphiteNiFe2.50.1 M NaOH5.41 × 10^−5^2.31 × 10^−4^1.80 × 10^−4^133245This article1.0 M NaOH--1.35 × 10^−4^110203Expanded graphiteNiFe5.00.1 M NaOH3.10 × 10^−5^2.32 × 10^−4^1.03 × 10^−4^116252This article1.0 M NaOH--2.63 × 10^−4^123198Expanded graphiteNiFe7.50.1 M NaOH1.53 × 10^−5^2.34 × 10^−4^3.14 × 10^−5^77219This article1.0 M NaOH--1.40 × 10^−4^119222Unmodified compressed graphite None00.1 M NaOH3.12 × 10^−6^3.06 × 10^−6^5.00 × 10^−5^84-This article1.0 M NaOH--1.42 × 10^−8^116596Compressed graphiteNiFe0.50.1 M NaOH1.70 × 10^−4^4.31 × 10^−4^1.20 × 10^−4^110194This article1.0 M NaOH--1.13 × 10^−4^93199Compressed graphiteNiFe2.50.1 M NaOH7.77 × 10^−5^2.24 × 10^−4^1.30 × 10^−4^113-This article1.0 M NaOH--1.30 × 10^−4^96201Compressed graphiteNiFe5.00.1 M NaOH1.65 × 10^−5^3.48 × 10^−4^3.47 × 10^−5^94-This article1.0 M NaOH--4.50 × 10^−5^102198Compressed graphiteNiFe7.50.1 M NaOH1.19 × 10^−5^2.61 × 10^−4^3.49 × 10^−5^104-This article1.0 M NaOH--2.50 × 10^−5^101204Ni foamNiFeRu LDH1.21.0 M KOH---3129[24]Ni foamNiFe LDH1.21.0 M KOH---153269[24]Glassy carbonNi3FeN NPs0.351.0 M KOH---42158[26]Carbon clothNi-FeP/TiN-1.0 M KOH---7375[27]Ni foamNiFe LDH0.451.0 M KOH---120195[25]Ni foamNiFeIr LDH0.451.0 M KOH---5651[25]Ni foamNiFeRh LDH0.451.0 M KOH---2724[25]Carbon clothPtNi-Ninanoarray-0.1 M KOH---4238[28]

The findings presented in this work are consistent with a significant portion of the relevant literature on electrocatalysis, demonstrating that the NiFe-modified expanded graphite and compressed graphite electrodes exhibit competitive catalytic performance for the HER without necessitating complex preparation methods. The above method contrasts with some documented sophisticated approaches, such as the in situ growth of ultrafine PtNi nanoparticle-decorated Ni nanosheet arrays on carbon cloth [28]. Such methods, while yielding high-performance catalysts, often involve complex preparation steps that can limit their scalability and practical application.

However, it is important to acknowledge that there are instances within the literature where certain catalysts exhibit superior performance metrics. These standout examples often share a common factor: the surface area used for calculating performance results is sometimes questionable or it is not directly comparable with standard measurements. This discrepancy could lead to challenges in directly comparing the efficiency of these advanced materials with those prepared using uncomplicated procedures.

In essence, while the latter approach may not match the peak performance of the most sophisticated catalysts in every instance, it offers a valuable balance between ease of preparation, cost-effectiveness, and competitive catalytic activity. This balance is crucial for the advancement of practical electrocatalytic applications, where scalability and accessibility of preparation techniques are as important as the ultimate efficiency of the catalyst.

#### 3.2.4. AC Impedance—OER

The EIS results for the expanded graphite and compressed graphite electrodes, both in their unmodified form and activated with various loadings of NiFe (0.5, 2.5, 5.0, and 7.5 mg cm^−2^), are gathered in Table 4 (the full potential range is presented in Appendix A). With respect to the OER, Figure 10 elucidates that the incorporation of NiFe modifications onto the base graphite electrodes significantly elevates their electrocatalytic reactivity.

Across the range of potentials tested for the unmodified expanded graphite electrodes, the *R*_p_ and *C*_p_ parameters demonstrated variability, lacking a definitive pattern, yet generally showing a tendency to decrease with an increase in electrode potential. Further insights into the behavior of these parameters and their implications for electrochemical performance were elaborated upon in Section 3.2.2, providing a deeper understanding of how the potential variations influence electrode dynamics. The catalytic OER enhancement due to NiFe deposition was pronounced for the expanded graphite electrode, with the disappearance of a semicircle related to the *R*_p_ and *C*_p_ parameters. Notably, the introduction of the NiFe particles resulted in an *R*_ct_ reduction by approximately 60 times and a *C*_dl_ increase by about 2 times at the potential of 1600 mV. Similarly to the HER findings, the catalytic properties for the OER on the expanded graphite were significantly augmented by NiFe deposition rather than a mere increase in the electrochemically active surface area.

In contrast, the compressed graphite electrode did not display the semicircle typically indicative of a porosity response, and subsequent NiFe modifications left this feature unchanged. Modification with the NiFe alloy led to a notable improvement in the OER activity, as evidenced by a reduction in the *R*_ct_ parameter by approximately 100 times at the potential of 1600 mV, with significant alterations in the *C*_dl_ parameter by ca. 39 times. Such radical changes in both the *R*_ct_ and *C*_dl_ parameters underscore the effectiveness of the NiFe alloy in optimizing the electrode’s surface catalytic properties for the OER, effectively enhancing its catalytic activity and implying improved alignment of the electrochemical properties with the desired reaction dynamics.

For all NiFe-modified electrodes (using the same base, with different loadings), an observable reduction trend in the reaction resistance with rising overpotential values could be noted, particularly for the lowest NiFe loading on the expanded graphite, where the resistance diminished from 41.1 to 3.4 Ω cm^2^, while for analogous NiFe loading on the compressed graphite samples, the *R*_ct_ reduced from 32.2 to 4.3 Ω cm^2^ for the potential range of 1500–1800 mV, respectively. Unlike in the HER scenario, the *C*_dl_ parameter for the OER showed an unspecific fluctuation with rising electrode potential for the NiFe-modified expanded graphite and compressed graphite-based electrodes, most likely indicating the significant impact of O_2_ bubble formation on the electrode surface (Table 4 and Appendix A).

It was observed that while an initial NiFe loading of 0.5 mg cm^−2^ significantly enhanced the electrochemical behavior, as demonstrated by decreased *R*_ct_ and increased *C*_dl_ parameter values, further increases in the NiFe loading did not continue to enhance their OER performance. Despite the anticipated benefits of having increased the catalyst presence, the performance reached a plateau, suggesting that beyond a certain threshold, additional NiFe mass does not contribute to further catalytic efficiency. This outcome suggests that, similarly to the HER behavior, there is an optimal NiFe loading level for maximizing the OER activity on expanded graphite electrodes beyond which the benefits diminish, possibly due to such factors as catalyst agglomeration or the obstruction of active sites, which counteract the potential advantages of having a larger catalytic surface area.

#### 3.2.5. Polarization Technique—OER

The Tafel polarization curves for both the unmodified and NiFe-modified expanded graphite and compressed graphite electrodes are illustrated in Figure 11a,b, demonstrating the electrocatalytic OER performance enhancements brought about by NiFe surface modification. The analysis of these curves within the 1500–1600 mV potential range provides a detailed view of the electrodes’ performance, with anodic slope (*b*_a_) and current density at an overpotential of 0.3 V (*j*_(ŋ = 0.3 V)_), as presented in Table 5 for comprehensive analysis.

Further highlighting the catalytic improvements, Figure 12a,b demonstrates that the NiFe-modified samples achieve significantly lower OER onset potential compared to their base material counterparts, signifying an earlier initiation of the oxygen evolution process. This performance enhancement positions the NiFe-modified electrodes on a par with or superior to those utilizing noble metal catalysts such as platinum, ruthenium, and iridium, as evidenced by the data recorded via the comparative analysis provided in Table 5. Table 5 highlights the effectiveness of NiFe as a catalyst and showcases a variety of transition metal combinations with comparable catalytic properties. This revelation significantly expands the pool of potential catalysts available for such applications and provides an opportunity to uncover more efficient and cost-effective catalytic materials that can be tailored to meet the demands of this important electrochemical process.

The recorded current density values (Table 5) at an anodic overpotential of 300 mV closely align with those of bulk NiFe-layered double hydroxide (LDH) materials, emphasizing the competitive nature of the NiFe modifications. By achieving even higher current densities than on such materials as IrO_2_ or CoP (see Table 5), the NiFe-modified expanded graphite and compressed graphite electrodes stand out for their superior electrocatalytic performance and their simplicity and scalability potential. This performance underscores the value of further research aimed at optimizing these materials, especially for enhancing water electrolysis technology.

Moreover, the overpotentials recorded at a current density of 10 mA cm^−2^ for the NiFe-modified electrodes with various NiFe alloy loadings underscore their efficacy and potential superiority when compared with other catalysts, as detailed in Table 5. These findings collectively emphasize the significant role of the NiFe modifications in advancing the electrocatalytic performance of graphite-based electrodes for OER applications, balancing between innovative catalytic efficiency and the practicality of their broader industrial use.

All NiFe-modified electrodes exhibit similar polarization behavior across the different electrolyte concentrations. However, there are noticeable differences among specific NiFe loadings; a clear trend shows that higher loadings initiate passivation at lower potentials. For instance, passivation starts at approximately 1.35 V for NiFe loadings of 0.5 mg cm^−2^, whereas for loadings of 7.5 mg cm^−2^, it begins around 1.15 V. This passivation process involves the formation of a protective Ni(OH)_2_ and Fe(OH)_3_ layer on the nickel-iron surfaces through a series of electrochemical reactions, leading to the development of this insoluble film [29,30]. As the potential increases beyond about 1.4 V, the formation of the β-NiOOH film commences and lasts until it reaches about 1.5 V. In the so-called “transpassive” region, beyond 1.5 V, the anodic current density dramatically increases. This phenomenon indicates a breakdown of the passivation layer, which may similarly affect the NiFe-modified electrodes, particularly given the similarity in chemical behavior that is imparted by the nickel component [31]. Thus, the long-term stability and corrosion resistance of these NiFe-modified electrodes under high potential operation in the transpassive region remain subjects of concern.
molecules-29-04755-t005_Table 5Table 5OER kinetic parameters for the selected catalyst materials.Base MaterialCatalystLoading[mg cm^−2^]ElectrolyteTafelRefs.*j*_0.3V_ [A cm^−2^]*b*_a_ [mV dec^−1^]*η*_(*j* = 10 mA cm^−2^)_ [mV]Pure expanded graphiteNone00.1 M NaOH6.9 × 10^−5^111-This article1.0 M NaOH1.3 × 10^−4^79-Expanded graphiteNiFe0.50.1 M NaOH4.2 × 10^−3^71-This article1.0 M NaOH1.7 × 10^−2^55286Expanded graphiteNiFe2.50.1 M NaOH2.8 × 10^−3^68-This article1.0 M NaOH1.4 × 10^−2^83270Expanded graphiteNiFe5.00.1 M NaOH2.3 × 10^−3^77-This article1.0 M NaOH2.2 × 10^−2^71278Expanded graphiteNiFe7.50.1 M NaOH1.7 × 10^−3^91-This article1.0 M NaOH2.2 × 10^−2^66282Pure compressed graphite None00.1 M NaOH2.4 × 10^−4^126-This article1.0 M NaOH1.0 × 10^−4^124-Compressed graphite NiFe0.50.1 M NaOH7.3 × 10^−3^55-This article1.0 M NaOH2.7 × 10^−2^36260Compressed graphite NiFe2.50.1 M NaOH2.4 × 10^−3^59-This article1.0 M NaOH2.6 × 10^−2^56277Compressed graphite NiFe5.00.1 M NaOH6.7 × 140^−3^40-This article1.0 M NaOH6.4 × 10^−3^52310Compressed graphite NiFe7.50.1 M NaOH1.3 × 10^−3^65-This article1.0 M NaOH1.0 × 10^−2^54300Fluorine-doped tin oxide glassNi_3_Co_3_Fe_3_ LDH-1.0 M NaOH-65290[32]Ni foamNiCo LDH1.760.1 M KOH-113290[33]Glassy carbonO-NiCoFe LDH0.120.1 M KOH--420[34]Ni foamNiFe LDH/N-rOG0.360.1 M KOH-63258[35]Carbon fiber clothNiCoFe LDH0.41.0 M KOH-32239[36]Ni foamNi_3_FeAl_0.91_ LDH0.51.0 M KOH-57304[37]Ni foamNiFeRu LDH1.21.0 M KOH-32225[24]Ni foamNiFe LDH1.21.0 M KOH--230[24]Carbon paperNiFe LDH-UF0.351.0 M KOH-32254[38]Glassy carbonIrO_2_0.711.0 M KOH
76340[39]Glassy carbonCoP NP5.01.0 M KOH
99340[39]Glassy carbonRuO_2_/CeO_2_-0.1 M NaOH1.0 × 10^−3^44-[40]-RuCu nanosheets-1.0 M KOH--234[41]

## 4. Conclusions

In conclusion, this investigation into the NiFe alloy modifications of expanded graphite and compressed graphite electrodes unveils a significant advancement in the realm of electrocatalysis, specifically for applications within alkaline water electrolyzers. The deposition of NiFe at various loadings markedly enhances the electrochemical performance of the HER and OER. This is evidenced by the notable reduction of the charge transfer resistance parameter, which indicates a more efficient electrochemical process under operational conditions.

Crucially, this study highlights an optimal threshold for NiFe loading (namely, 0.5 mg cm^−2^), beyond which any additional catalyst material does not yield proportional gains in electrode performance, potentially due to such factors as catalyst agglomeration or active site saturation. The exploration of transition metal combinations further broadens the horizons for developing more efficient, cost-effective catalysts that are tailored to specific electrochemical processes, thus opening a rich field for future research and development.

While these NiFe-modified graphite electrodes demonstrate promising results, surpassing even some noble metal counterparts in specific metrics, a clear avenue still remains for their catalytic optimization. Enhancing the HER activity and fine-tuning the material’s composition and electrochemical properties will be essential to realizing a highly efficient, commercially viable alkaline water electrolysis system. The pursuit of green hydrogen production as a clean energy carrier underscores the importance of such advancements, driving toward sustainable and economically feasible solutions in the energy sector.

Thus, the next stage should involve constructing an electrolyzer featuring a stack using such advanced electrodes, a critical step that is outlined for forthcoming research endeavors. This initiative is integral to transitioning from laboratory discoveries of tangible, scalable solutions in the realm of green hydrogen production, aligning with broader objectives to foster sustainable and economically viable energy alternatives. These efforts, planned for future works, will be instrumental in evaluating the practical efficacy, durability, and overall system efficiency, along with a comprehensive understanding of the potential uses for these electrodes in real-world applications.

## Figures and Tables

**Figure 1 molecules-29-04755-f001:**
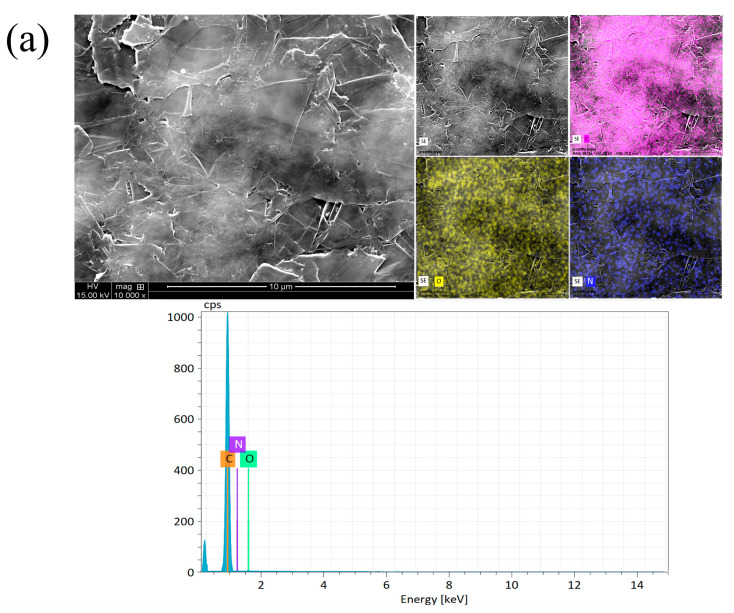
SEM micrograph pictures of expanded graphite, taken at 10,000× magnification (top left); EDX elemental mappings (top right) with the maps color-coded as follows: pink for carbon, yellow for oxygen, blue for nitrogen, purple for nickel, and green for iron, while areas where green and purple overlap indicate the presence of both iron and nickel; EDX pattern (bottom) for the unmodified electrode (**a**) and for all NiFe loadings, namely: 0.5 (**b**), 2.5 (**c**), 5.0 (**d**), and 7.5 (**e**) mg of NiFe per cm^2^.

**Figure 2 molecules-29-04755-f002:**
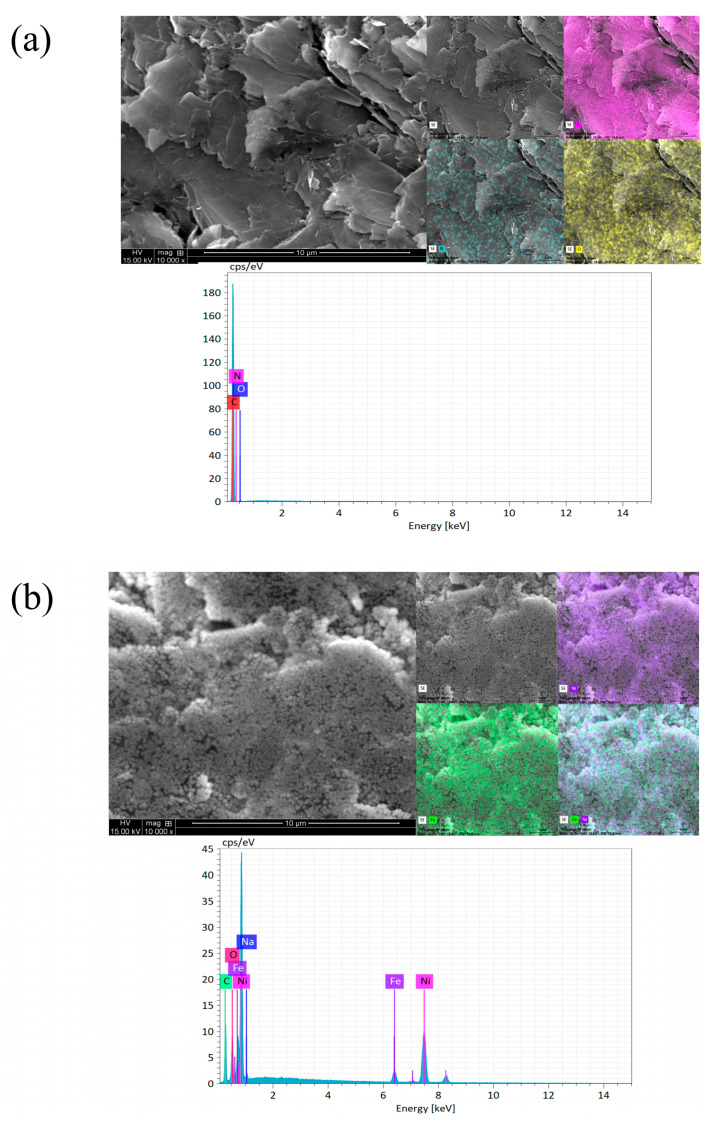
SEM micrograph pictures of compressed graphite, taken at 10,000× magnification (top left); EDX elemental mappings (top right), with the maps color-coded as follows: pink for carbon, yellow for oxygen, blue for nitrogen, purple for nickel, and green for iron, while areas where green and purple overlap indicate the presence of both iron and nickel. EDX pattern (bottom) for the unmodified electrode (**a**) and for all NiFe loadings, namely: 0.5 (**b**), 2.5 (**c**), 5.0 (**d**), and 7.5 (**e**) mg of NiFe per cm^2^.

**Figure 3 molecules-29-04755-f003:**
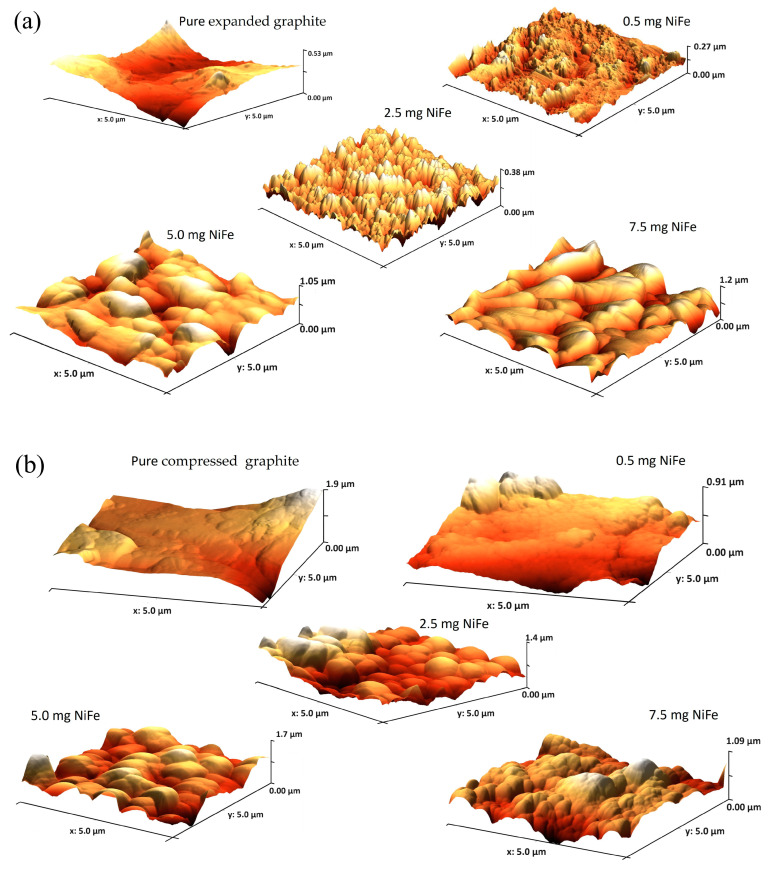
AFM images comparing unmodified and NiFe-modified graphite electrodes for (**a**) expanded graphite and (**b**) compressed graphite.

**Figure 4 molecules-29-04755-f004:**
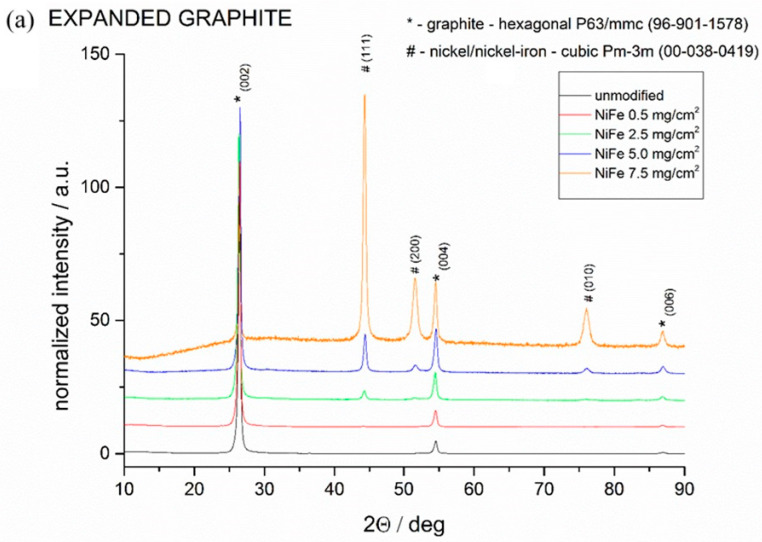
XRD patterns of unmodified and NiFe-modified graphite electrodes for (**a**) expanded graphite and (**b**) compressed graphite.

**Figure 5 molecules-29-04755-f005:**
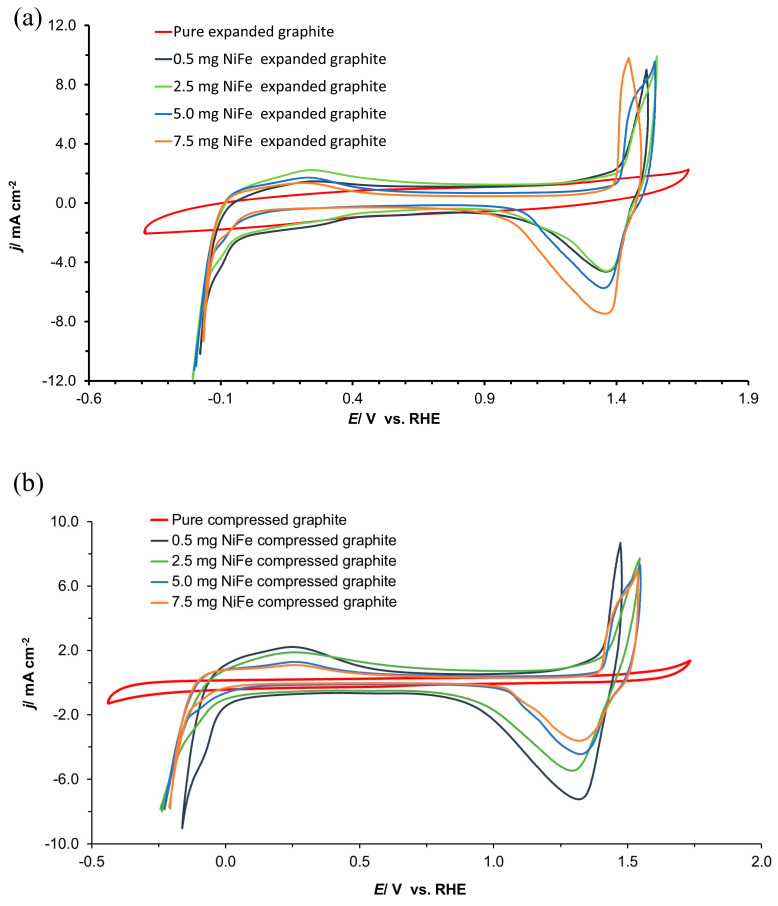
Cyclic voltammogram curves of (**a**) expanded graphite and NiFe-modified expanded graphite; (**b**) compressed graphite and NiFe-modified compressed graphite electrodes in contact with 0.1 M NaOH medium, carried out at a scan rate of 50 mV s^−1^ over the potential span of −0.5 to 1.8 V vs. RHE for the indicated NiFe loading levels.

**Figure 6 molecules-29-04755-f006:**
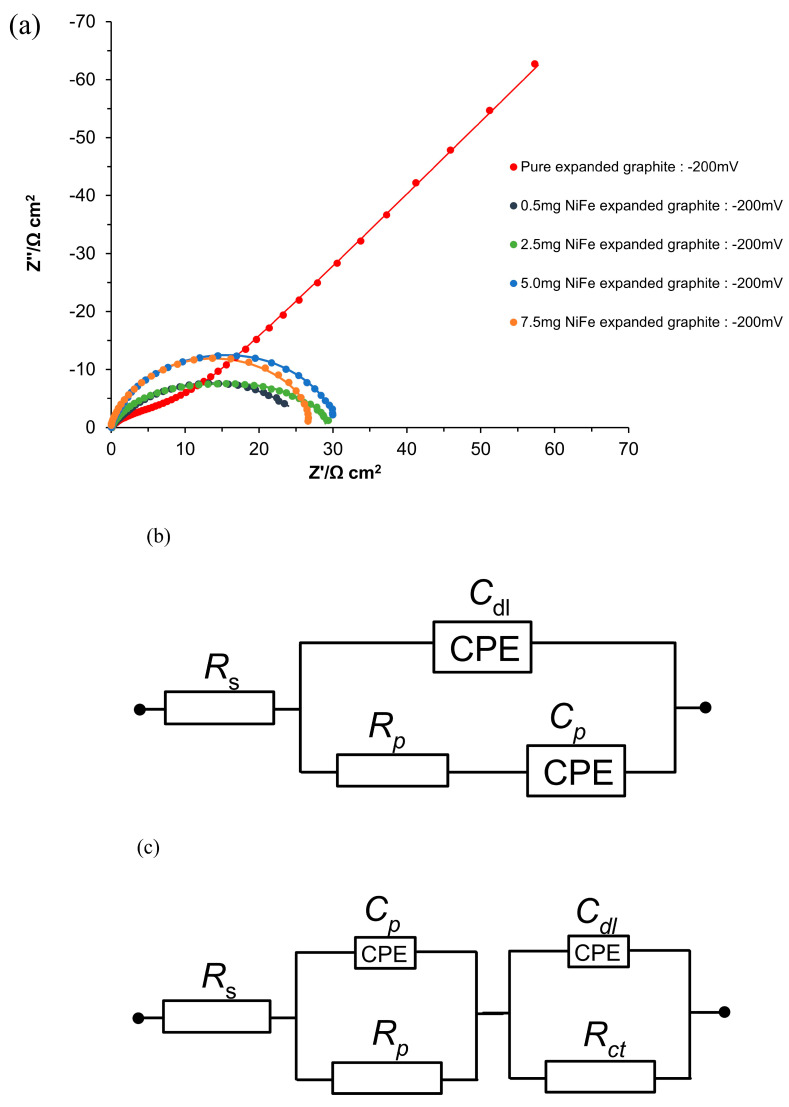
(**a**) Electrochemical Nyquist impedance plots for the HER on unmodified and NiFe-modified electrode surfaces in 0.1 M NaOH for the potential of −200 mV vs. RHE; (**b**–**d**) equivalent circuits used to fit the above processes, where *C*_p_ is the Faradaic pseudocapacitance, *R*_p_ is the Faradaic resistance, and *C*_dl_ is the double-layer capacitance (both capacitance parameters are CPE or constant phase element−modified), jointly in series with an uncompensated solution resistance, *R*_s_. The data derived from the equivalent circuits are represented by the solid lines.

**Figure 7 molecules-29-04755-f007:**
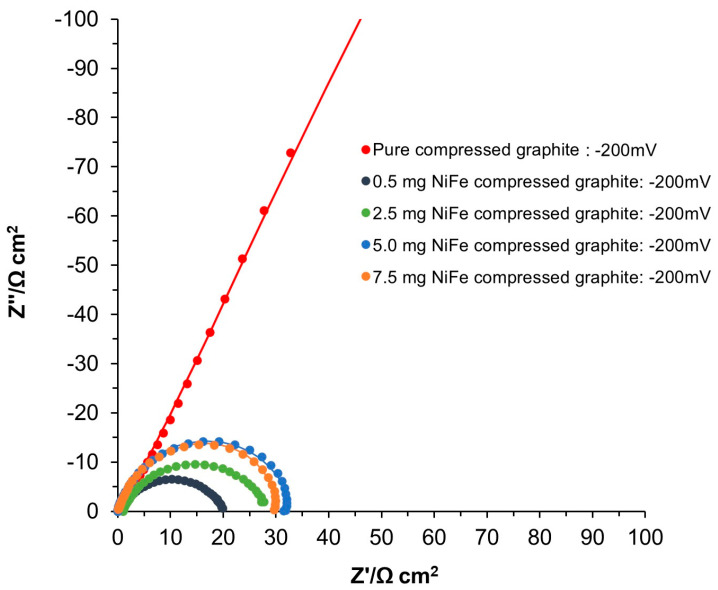
Electrochemical Nyquist impedance plots for the HER on unmodified and NiFe-modified electrode surfaces in 0.1 M NaOH for the potential of −200 mV vs. RHE.

**Figure 8 molecules-29-04755-f008:**
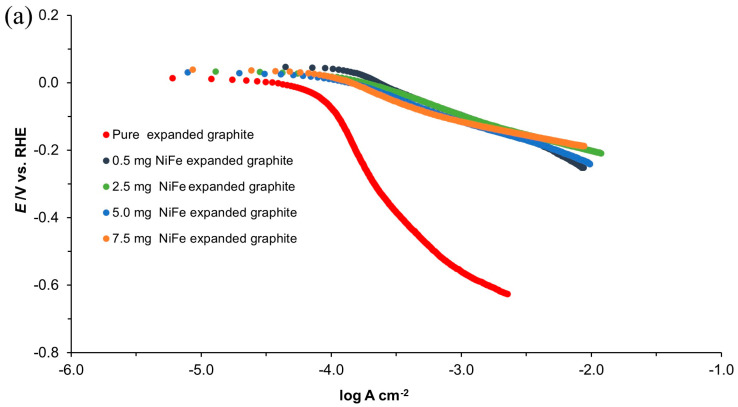
Quasi-potentiostatic cathodic Tafel polarization curves for the HER, obtained from the expanded graphite (**a**) and compressed graphite (**b**) electrodes, both unmodified and modified, with various NiFe loadings in 0.1 M NaOH electrolyte. The curves were recorded at a scan rate of 0.5 mVs^−1^ (iR-corrected).

**Figure 9 molecules-29-04755-f009:**
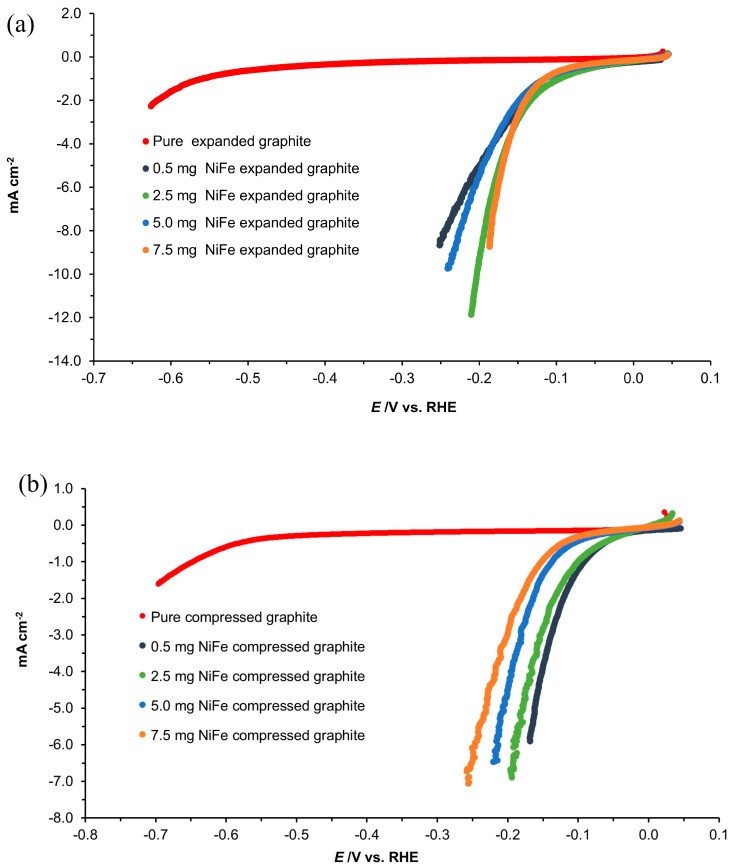
Linear sweep voltammetry (LSV) curves of expanded graphite (**a**) and compressed graphite (**b**) electrodes, both unmodified and modified, with various NiFe loadings in 0.1 M NaOH solution, carried out with a scan rate of 0.5 mV s^−1^ for the HER (iR-corrected).

**Figure 10 molecules-29-04755-f010:**
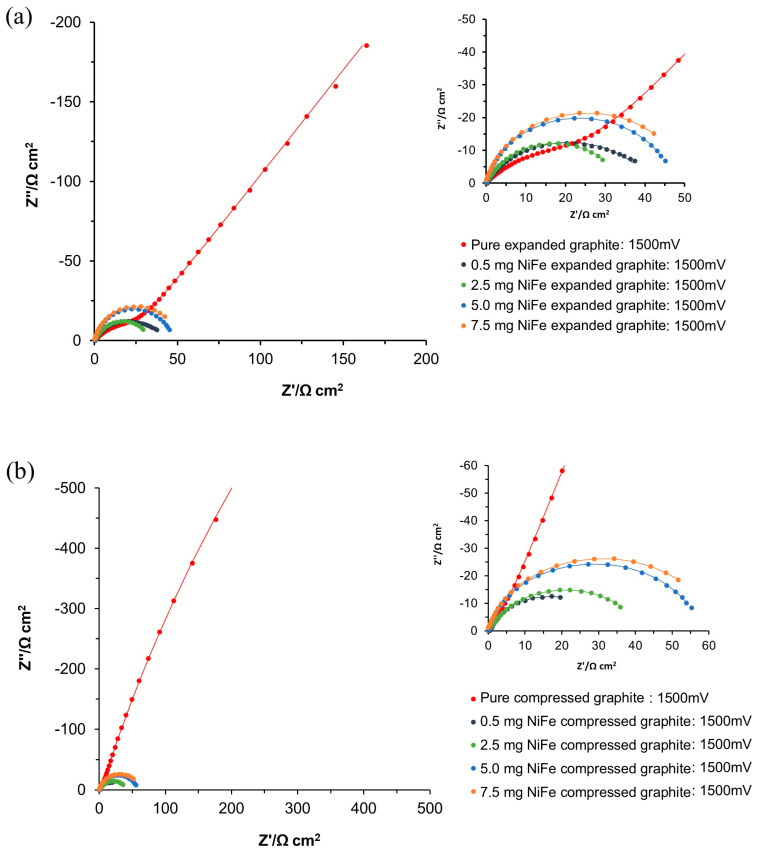
Electrochemical Nyquist impedance plots for the OER on expanded graphite (**a**) and compressed graphite (**b**) and their NiFe-modified electrode surfaces in contact with 0.1 M NaOH solution for a potential of 1500 mV vs. RHE.

**Figure 11 molecules-29-04755-f011:**
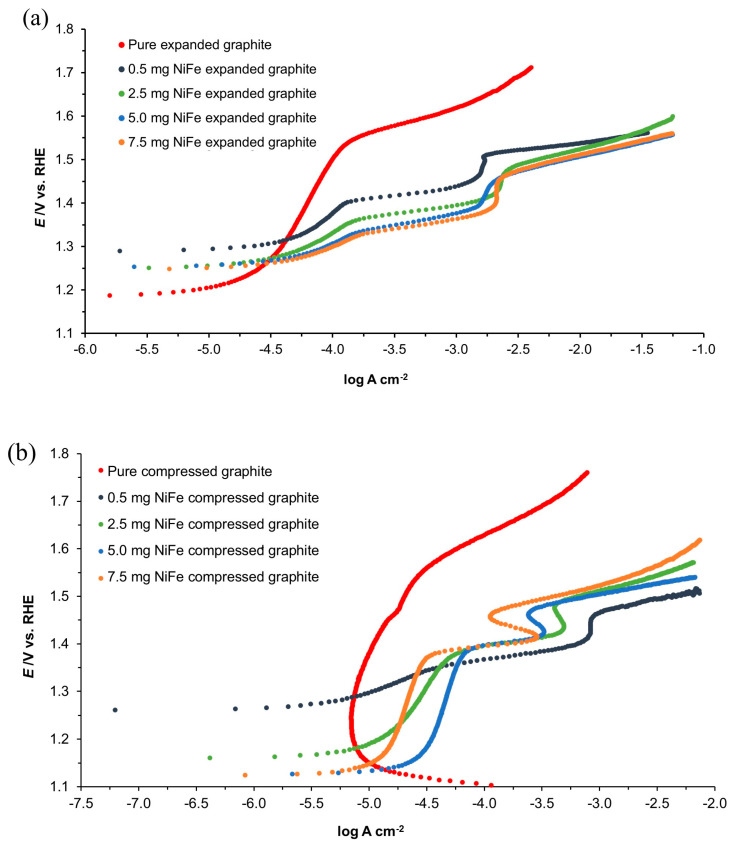
Quasi-potentiostatic cathodic Tafel polarization curves for the OER, obtained for the expanded graphite (**a**) and compressed graphite (**b**) electrodes, both unmodified and modified with various NiFe loadings in 0.1 M of NaOH electrolyte. The curves were recorded at a scan rate of 0.5 mVs^−1^ (iR-corrected).

**Figure 12 molecules-29-04755-f012:**
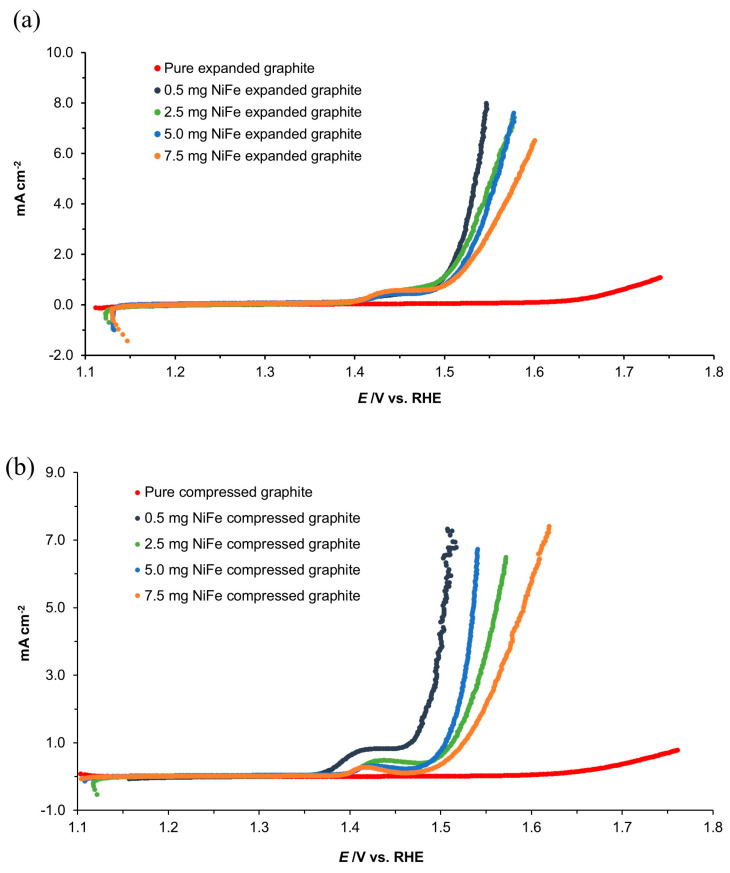
Linear sweep voltammetry (LSV) curves of expanded graphite (**a**) and compressed graphite (**b**) electrodes, both unmodified and modified with various NiFe loadings in 0.1 M of NaOH solution, carried out with a scan rate of 0.5 mV s^−1^ for the OER (iR-corrected).

**Table 1 molecules-29-04755-t001:** Double−layer capacitance (*C*_dl_) values for expanded graphite and compressed graphite electrodes with varying NiFe loadings (the data are relevant to Appendix A).

	Expanded Graphite					Compressed Graphite				
Loading [mg cm^−2^]	0	0.5	2.5	5.0	7.5	0	0.5	2.5	5.0	7.5
Capacitance [uF]	9000	7700	4500	3600	2500	2900	22400	7500	7300	3500

**Table 2 molecules-29-04755-t002:** Electrochemical impedance parameters for the HER obtained for the expanded graphite and compressed graphite electrodes, both unmodified and modified with various NiFe loadings in 0.1 M NaOH supporting solution. The results obtained here were recorded by fitting the CPE-modified Randles equivalent circuits (see Figure 6b–d) to the experimentally obtained impedance data (reproducibility is usually below 10%, χ^2^ = 1.56 × 10^−6^ to 1.74 × 10^−5^).

*E*/mV	*R*_p_/Ω cm^2^	*C*_p_/µF cm^−2^	*R*_ct_/Ω cm^2^	*C*_dl_/µF cm^−2^
**Unmodified** **expanded graphite**				
−50	17.3 ± 1.1	3495 ± 346	-	7999 ± 380
−400	28.3 ± 1.1	7863 ± 143	-	4693 ± 152
−450	10.5 ± 0.3	4447 ± 320	396.2 ± 7.4	12,351 ± 35
−700	7.1 ± 0.5	1960 ± 298	24.5 ± 0.8	12,815 ± 290
**NiFe 0.5 mg cm^−2^**				
−50	51.2 ± 3.3	11,879 ± 237	123.1 ± 0.7	13,277 ± 185
−450	-	-	6.1 ± 0.1	3474 ± 241
**NiFe 2.5 mg cm^−2^**				
−50	22.2 ± 3.2	9257 ± 436	123.8 ± 1.4	10,673 ± 173
−450	-	-	4.7 ± 0.0	1948 ± 43
**NiFe 5.0 mg cm^−2^**				
−50	170.6 ± 0.8	3970 ± 155	178.0 ± 24.6	3022 ± 168
−450	-	-	4.4 ± 0.1	848 ± 56
**NiFe 7.5 mg cm^−2^**				
−50	238.0 ± 0.9	2205 ± 37	276.5 ± 4.6	1360 ± 16
−450	-	-	5.7 ± 0.1	464 ± 27
**Unmodified** **compressed graphite**				
−50	4.3 ± 0.2	9783 ± 688	2162.9 ± 19.1	2766 ± 4
−450	1.4 ± 0.2	4066 ± 772	494.3 ± 5.9	3763 ± 16
−650	38.2 ± 3.6	7097 ± 200	140.8 ± 6.9	20,992 ± 2072
−700	22.4 ± 6.8	5242 ± 464	51.7 ± 5.4	40,964 ± 3549
**NiFe 0.5 mg cm^−2^**				
−50	3.6 ± 0.8	3923 ± 1150	39.1 ± 1.9	9741 ± 110
−450	-	-	5.7 ± 0.1	5501 ± 172
**NiFe 2.5 mg cm^−2^**				
−50	-	-	97.5 ± 0.9	3154 ± 45
−450	-	-	6.7 ± 0.1	1457 ± 99
**NiFe 5.0 mg cm^−2^**				
−50	-	-	318.7 ± 1.9	976 ± 7
−450	-	-	8.2 ± 0.9	676 ± 60
**NiFe 7.5 mg cm^−2^**				
−50	-	-	362.8 ± 3.8	841 ± 9
−450	-	-	5.7 ± 0.1	417 ± 25

**Table 4 molecules-29-04755-t004:** Electrochemical parameters for the OER obtained for the expanded graphite and compressed graphite electrodes, both unmodified and modified with various NiFe loading levels in 0.1 M of NaOH supporting solution. The results obtained here were recorded by fitting the CPE-modified Randles equivalent circuits (see Figure 6b–d) to the experimentally obtained impedance data (reproducibility is usually below 10%, χ^2^ = 1.56 × 10^−6^ to 1.74 × 10^−5^).

*E*/mV	*R*_p_/Ω cm^2^	*C*_p_/µF cm^−2^	*R*_ct_/Ω cm^2^	*C*_dl_/µF cm^−2^
**Unmodified expanded graphite**				
1300	72.5 ± 4.2	1427 ± 129	-	4762 ± 153
1400	50.2 ± 2.2	2302 ± 144	-	5379 ± 173
1600	44.9 ± 3.5	3103 ± 247	523.2 ± 80.7	8816 ± 108
1800	36.4 ± 1.7	906 ± 27	53.0 ± 3.5	11,069 ± 847
**NiFe 0.5 mg cm^−2^**				
1300	35.3 ± 3.3	3606 ± 301	-	14,140 ± 195
1400	-	-	351.9 ± 26.3	31,733 ± 188
1500			41.1 ± 0.1	20,773 ± 51
1600	-	-	8.6 ± 0.1	16,106 ± 238
1800			3.4 ± 0.2	19,283 ± 2338
**NiFe 2.5 mg cm^−2^**				
1300	33.0 ± 2.1	4241 ± 230	-	9763 ± 166
1400	-	-	2427.2 ± 469.4	27,367 ± 81
1500			33.6 ± 0.1	45,891 ± 160
1600	-	-	6.8 ± 0.1	44,530 ± 873
1800			2.2 ± 0.1	34,172 ± 3782
**NiFe 5.0 mg cm^−2^**				
1300	489.1 ± 15.4	2594 ± 18	-	4710 ± 148
1400	-	-	874.4 ± 23.3	9843 ± 39
1500			47.1 ± 0.1	22,100 ± 32
1600	-	-	8.5 ± 0.0	20,093 ± 238
1800			2.8 ± 0.1	4710 ± 148
**NiFe 7.5 mg cm^−2^**				
1300	305.5 ± 7.8	2362 ± 15	-	4038 ± 54
1400	-	-	506.6 ± 15.5	18,150 ± 89
1500			51.3 ± 0.1	21,468 ± 42
1600	-	-	7.7 ± 0.0	21,551 ± 143
1800			2.5 ± 0.2	18,765 ± 589
**Unmodified compressed graphite**				
1300	-	-	-	1607 ± 6
1400	-	-	-	1657 ± 7
1500	-	-	6020.1 ± 227.8	1703 ± 6
1600	-	-	942.9 ± 9.7	1651 ± 9
1800	-	-	101.0 ± 0.5	1498 ± 19
**NiFe 0.5 mg cm^−2^**				
1300	46.5 ± 6.2	11,402 ± 303	-	3527 ± 314
1400	2.1 ± 0.2	31,534 ± 2106	-	30,076 ± 2104
1500	0.5 ± 0.0	62,836 ± 8838	32.2 ± 0.2	66,025 ± 188
1600	-	-	9.5 ± 0.1	63,908 ± 453
1800	-	-	4.3 ± 0.1	67,769 ± 2389
**NiFe 2.5 mg cm^−2^**				
1300	248.2 ± 53.6	7532 ± 279	-	-
1400	8.9 ± 0.9	6037 ± 788	-	-
1500	-	-	34.1 ± 0.1	10,430 ± 335
1600	-	-	8.3 ± 0.0	50,340 ± 178
1800	-	-	3.4 ± 0.1	52,233 ± 462
**NiFe 5.0 mg cm^−2^**				
1300	397.5 ± 17	3719 ± 28	-	3395 ± 110
1400	132.9 ± 9.8	30,423 ± 303	-	15,350 ± 45
1500	-	-	32.8 ± 0.1	26,630 ± 93
1600	-	-	7.1 ± 0.0	29,546 ± 359
1800	-	-	2.5 ± 0.1	30,010 ± 991
**NiFe 7.5 mg cm^−2^**				
1300	-	-	-	2826 ± 11
1400	-	-	-	26,412 ± 133
1500	-	-	65.3 ± 0.3	19,398 ± 64
1600	-	-	8.3 ± 0.1	21,559 ± 456
1800	-	-	2.4 ± 0.1	16,191 ± 2277

## Data Availability

Dataset available on request from the authors.

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
