# Peer review of "Optimizing NiFe-Modified Graphite for Enhanced Catalytic Performance in Alkaline Water Electrolysis: Influence of Substrate Geometry and Catalyst Loading"

_molecules, 2024, doi:10.3390/molecules29194755_

Round 1
Reviewer 1 Report
Comments and Suggestions for Authors
In this manuscript, the authors provided a thorough study on the effect of substrate geometry and catalyst loading on the oxygen evolution reaction (OER) and hydrogen evolution reaction (HER) performances of NiFe catalysts. This can have implications for understanding the interaction between the catalyst and the substrate and how this impacts the overall efficiency of alkaline water electrolysis. Overall, this is an interesting contribution to the OER/HER field. Therefore, I would like to express my strong support for the publication of this work at Molecules. However, some minor concerns are found and need to be properly addressed in order to further improve the clarity and quality of the manuscript. The authors might find the below detailed comments useful.
1. Did the authors apply iR-correction to the LSV data? Some of the data might have been overly iR-compensated, for example, the blue curve (5.0 mg NiFe Compressed Graphite) in Figure 12b. Please refer to this commentary for iR compensation in electrocatalysis research (DOI: 10.1021/acsenergylett.3c00366).
2. The double-layer capacitance was assessed by running CV measurements at varying scan rates from 5 to 100 mV s-1. It is suggested that the authors provide these CV data in the Supporting Information.
3. Recent works on alkaline water splitting and be referenced in the Introduction (e.g., DOI: 10.1002/inf2.12608; 10.1016/j.matre.2022.100144).
4. Figures 1 and 2 extensively presented the SEM and mapping images of all the control samples. Maybe consider presenting only those of the representative samples and move the rest to the Supporting Information.
5. The authors only briefly mentioned the electrodeposition of the NiFe catalysts by simply referring to published results. Some description is still needed to help readers understand the methodology. What are the atomic ratio between Ni and Fe?
6. The two kinds of graphite seem to have different characteristic peaks in the XRD data. Why? Please provide some discussion.
Reviewer 2 Report
Comments and Suggestions for Authors
The authors present a detailed study on NiFe-modified graphite electrodes, focusing on the influence of substrate geometry and catalyst loading on the performance of water electrolysis in alkaline conditions. The study compares the effects of compressed and expanded graphite substrates, demonstrating that substrate geometry plays a significant role in catalytic efficiency. The paper provides comprehensive experimental data, including electrochemical and surface characterization techniques, to investigate how different NiFe loadings impact the electrodes' performance in the hydrogen evolution reaction (HER) and oxygen evolution reaction (OER). However, there are some aspects where revisions are recommended:
1. The authors mention that increased NiFe loading results in agglomeration and reduced catalytic efficiency. Could the authors provide more detailed analysis on how agglomeration specifically impacts electron transfer and active site accessibility? Perhaps including particle size distribution analysis for various loadings would add clarity.
2. The study compares expanded and compressed graphite but does not thoroughly explain why compressed graphite, despite having a larger surface area, showed lower activity. Could the authors delve deeper into this contradiction, possibly exploring differences in electrical conductivity or contact resistance between the substrate and catalyst?
3. Several works are reported by different groups, which could be mentioned, such as Adv. Mater. 2023, 2305074; Carbon Energy. 2024, e465.
4. The reduction in charge transfer resistance (Rct) upon NiFe modification is an important finding. Could the authors expand on how different NiFe loadings influence the Rct values in more detail, and how this correlates with the electrochemical activity observed in the cyclic voltammetry (CV) tests?
